# A full formal representation of Arrow's impossibility theorem

Kazuya Yamamoto *

Department of International Relations, Kyoto Sangyo University, Kyoto, Japan

* yamamoto@cc.kyoto-su.ac.jp

## Abstract

Revised proofs of Kenneth Arrow's impossibility theorem, one of the most influential theorems in economics, political science, and philosophy, have been presented in prose form, incorporating novel ideas such as decisive sets and pivotal voters. This study develops another approach to proving the theorem. Using a proof calculus in formal logic, we construct a proof with a full mathematical representation. While previous proofs emphasize intuitive accessibility, this one focuses on meticulous derivation and reveals the global structure of the social welfare function central to the theorem. The primary aim is to contribute methodologically to research on the theorem by demonstrating the effectiveness of systematically applying techniques from formal logic to its proof. Additionally, it accommodates a broader range of preference relations than those typically considered rational in standard economic models, allowing for the integration of diverse human behavior patterns into a single theoretical framework. The interdisciplinary relevance of the theorem is also discussed, including its relation to epistemology and philosophy.

## 1. Introduction

"… [I]t [Bertrand Russell's *Introduction to Mathematical Philosophy*] made a tremendous impression on me" [1, p. 43]. This is Kenneth Arrow's reminiscence about his high school days, as described in an interview conducted by Jerry Kelly. While logician Alfred Tarski's influence on Arrow during his undergraduate studies is well known [2], the above remark reveals that his interest in logic is much deeper. This mindset encouraged him to develop novel arguments based on relations in logic [3]. The (general) possibility theorem, also known as the impossibility theorem, was introduced in [4]. It gained widespread recognition after the publication of his book *Social Choice and Individual Values* the following year [5].

Despite Arrow's inspiration being drawn from formal logic, the proof of the theorem presented in his papers relied largely on words rather than formulas. It was a combination of logical expressions and verbal inferences, which may not have been entirely clear. This prompted a sequence of revised proofs. Arrow reorganized his proof

**Data availability statement:** All relevant data are within the manuscript and its Supporting information file.

**Funding:** The author(s) received no specific funding for this work.

**Competing interests:** The authors have declared that no competing interests exist.

immediately after publication [6]. Inada was the first scholar, other than Arrow, to present a new proof [7]. Blau identified minor mistakes in the original proofs and offered a modified version [8], which led Arrow to adjust the theorem's conditions and simplify the proof [9]. However, Sen criticized it as "opaque" [10, p.42] and sought to present a more reader-friendly proof by restating Arrow's proof in [9]. Subsequent proofs included [11–23], with Barberà's idea of pivotal voters being particularly noteworthy [24]. Geanakoplos developed a standard proof based on this idea (the first proof in [25]). Although it was simple and highly accessible to readers, it omitted non-trivial components, which could lead to misinterpretation. Nipkow and Wiedijk discussed the missing cases and ambiguity of statements in Geanakoplos's [25,26] proofs [27,28], and modified versions of the pivotal-voter proof were presented [29,30].

Deductive techniques in logic enable proofs constructed only by applying axioms and inference rules to logical formulas, and logic's advantage lies in such a rigorous procedure. In this light, the repeated reformulations of the proof, accompanied by verbal statements, likely deviated from the scholarly impact Arrow envisioned when he initially conceived of using the method. In fact, Arrow preferred proofs that were constructed by reducing verbal explanations to the minimum: "I [Arrow] was fascinated by this [the idea of logic] and used to aggravate my professors by writing out proofs in very strictly logical form, avoiding words as much as possible and things of that kind" [1, p. 44]. Arrow's choice to adopt a different approach would have been influenced by the prevailing state of the art at that time. As illustrated by Independence of Irrelevant Alternatives (IIA), a premise of the theorem, statements of the theorem require moving across profiles—sets of individual preference relations—in a social welfare function of the theorem, and the simplest and most natural approach to this property involves using higher-order logic (i.e., the first-order corresponds to a profile and the second-order corresponds to a set of profiles). However, higher-order logic was not fully developed at the time. Hilbert and Ackermann's seminal book, *Grundzüge der Theoretischen Logik* (Principles of Mathematical Logic), which discussed this topic [31], had already been published, but the English edition was not available until 1950, reflecting the research landscape of logic during the period [32].

This study returns to Arrow's initial inspiration. Using second-order logic, this work presents a proof that fully exploits the advantages of formal logic: a proof that derives the theorem's conclusion from its premises solely by manipulating logical formulas. Thus, we contribute to methodological development in social sciences by providing a new framework for mathematical economic analysis.

The proof of the impossibility theorem has engaged not only the economists mentioned above, but also computer scientists and logicians. Computer scientists aim to develop interactive theorem provers, which are used to verify proofs written by humans, and the proofs of the impossibility theorem, such as [9,10,25], are employed as a means of evaluating the performance of their provers [27,28,33] (see also [34]). The proofs are encoded in the provers' languages to align with higher-order logic, yet they manifest as computer source code specific to the provers' internal systems, thereby making them less accessible to those unfamiliar with the provers' specifications. Consequently, these are not conventional descriptions of mathematical proof

and, to my knowledge, the present proof is the first described in a standard form of second-order logic. Economists' proofs introduce concepts such as decisive sets and pivotal voters, while mechanized reasoning studies merely reconstruct the reasoning flow used in those proofs. Although this approach is valid for computer scientists evaluating provers' functionality, second-order logic allows for a proof that does not depend on such specialized concepts. This study offers a more straightforward proof that avoids these devices. Meanwhile, logicians aim to demonstrate the capabilities of specific logic classes or logic-based theories. Modal logic, a logic suitable for capturing necessity and possibility, is a prominent topic in logic, and those who aim at developing their original modal logic-based languages use the economists' proofs, or part of it, for their languages' benchmark testing [35–38]. Second-order arithmetic, an arithmetic theory based on second-order logic, is also successfully applied to proving the theorem using decisive sets [39].

The scholarship on the theorem extends well beyond its proof. Economic analyses have explored modified conditions, such as infinite populations and relaxed assumptions regarding preference ordering, with a comprehensive review available in [19]. Furthermore, the theorem intersects with various disciplines, including political science, game theory, physics, and engineering design [40–42]. Its impact also spans epistemology and philosophy, notably through a reformulation of Thomas Kuhn's theory choice argument via the impossibility theorem [43] and John Rawls' reconsideration of his theory of justice in light of Arrow's challenge [44,45].

Second-order logic is powerful enough to allow for the full formalization of the impossibility theorem, but it also has limitations; notably, unlike first-order logic, the completeness of proof calculi does not hold in second-order logic. This means some semantically valid inferences in so-called standard semantics cannot be proven by proof calculi in second-order logic. Although controversial, our intuitive sense of valid reasoning often aligns with semantic validity. Thus, the range of provable statements in second-order logic likely falls short of the inferences we intuitively deem valid. However, the soundness of second-order logic is guaranteed, indicating that any inference proven by proof calculi in second-order logic is also semantically valid; in this study, we establish the proof, thereby ensuring the semantic validity of the impossibility theorem. Section 3.2 illustrates this point.

After describing the argument of the impossibility theorem in the next section, the statements of the theorem are transformed into formulas in Section 3.1. Section 3.2 illustrates how the proof calculus employed in this study is carried out using a simple example. The proof is presented in section 4, each subsection beginning with a non-technical overview, followed by a more technical exposition grounded in logic. The concluding section discusses implications for economics and other disciplines.

## 2. The argument of the impossibility theorem

The impossibility theorem comprises two axioms—completeness and transitivity—and four conditions—unrestricted domain, unanimity, IIA, and non-dictatorship (it should be noted that the terms axioms and completeness are used in economics in a manner that differs from their usage in logic; in what follows, their meaning will vary depending on context) [46]. In a profile, every individual has a preference relation over alternatives and can have any preference that satisfies the two axioms. A profile is a tuple of individual preference relations. The set of profiles is unrestricted in the sense that it must contain all possible profiles [8]. A social welfare function defined on that set is a rule that assigns a social preference, which also satisfies the axioms, to each profile. Unanimity demands that if every individual strictly prefers an alternative to another in a profile, a social welfare function assigns the same social preference to the profile as that of the individuals. IIA demands that a social welfare function assigns the same social preference over two alternatives to those profiles among which every individual keeps their pairwise preference unchanged. A dictatorship is a social welfare function that has a single individual such that the function assigns the individual's preference to profiles whenever that person has a strict preference relation over two alternatives. The theorem argues that in a society where the number of alternatives is at least three, any social welfare function that satisfies the two axioms, unrestricted domain, unanimity, and IIA entails a dictatorship. (As the theorem holds trivially in the case of an individual, it usually assumes more than one individual).

## 3. Formalization of the theorem's statements and illustration of proof procedure

Section 3.1 translates the statements of the theorem into formulas. As some readers may be unfamiliar with proof theory in logic, section 3.2 outlines the procedure by which a proof is constructed using this technique, with a simple example.

### 3.1. Translation of the premises and conclusion into formulas

We have a language with the following translation keys:

$s$: society
$H(x)$: $x$ is an individual
$A(x)$: $x$ is an alternative
$R_n(w, x, y)$: $w$ weakly prefers $x$ to $y$ in profile $n$
$\textbf{P}(X)$: $X$ is a profile.

Predicate symbols and variables are denoted by capital letters. In our language, $H$, $A$, $R_n$, and $\textbf{P}$ are symbols, and $X$ is a variable. Bold letters are used for second-order predicates. $\textbf{P}$ is such a predicate in the language.

*The numbers of individuals and alternatives.* $I$ individuals exist in society, and they have $J$ alternatives:

$$\exists x_1 \ldots \exists x_I (H(x_1) \wedge \ldots \wedge H(x_I) \wedge x_1 \neq x_2 \ldots \wedge x_{I-1} \neq x_I \\ \wedge \forall y(H(y) \rightarrow (y = x_1 \vee \ldots \vee y = x_I))), \tag{1}$$

$$\exists x_1 \ldots \exists x_J (A(x_1) \wedge \ldots \wedge A(x_J) \wedge x_1 \neq x_2 \ldots \wedge x_{J-1} \neq x_J \\ \wedge \forall y(A(y) \rightarrow (y = x_1 \vee \ldots \vee y = x_J))), \tag{2}$$

where $I$ is equal to or greater than two, and $J$ is equal to or greater than three. Subformula $H(x_1) \wedge \ldots \wedge H(x_I)$ in (1) states that $I$ individuals exist. $x_1 \neq x_2 \wedge \ldots \wedge x_{I-1} \neq x_I$ states that all individuals are distinct. $\forall y(H(y) \rightarrow (y = x_1 \vee \ldots \vee y = x_I))$ states that no individual other than them exists. (2) has a composition similar to (1). Profiles are also written in a similar manner:

$$\exists X_1 \ldots \exists X_N (\textbf{P}(X_1) \wedge \ldots \wedge \textbf{P}(X_N) \wedge X_1 \neq X_2 \ldots \wedge X_{N-1} \neq X_N \\ \wedge \forall Y(\textbf{P}(Y) \rightarrow (Y = X_1 \vee \ldots \vee Y = X_N))), \tag{3}$$

where $N$ is the number of profiles. (Fishburn studied social welfare functions with the hypothesis that the set of individuals is infinite [47]).

*Unrestricted Domain* (*Universality*). For any two alternatives in any profile, every individual might have any pairwise preference relation that is logically possible:

$$\forall X(\textbf{P}(X) \rightarrow \forall w(H(w) \rightarrow \forall x \forall y((A(x) \wedge A(y)) \rightarrow \\ ((X(w, x, y) \vee \neg X(w, x, y)) \wedge (X(w, y, x) \vee \neg X(w, y, x)))))). \tag{4}$$

The domain stated in (4) includes (truly) all logically possible preference relations; it includes pairwise preference relations represented by $\neg X(w, x, y) \wedge \neg X(w, y, x)$. However, since completeness and transitivity are formulated in (6)–(9), and these formulas will be imposed on (4) in the deduction, the domain in the proof corresponds to the one normally assumed in the argument about the impossibility theorem. Social preference is also unrestricted:

$$\forall X(\textbf{P}(X) \rightarrow \forall x \forall y((A(x) \wedge A(y)) \rightarrow \\ ((X(s, x, y) \vee \neg X(s, x, y)) \wedge (X(s, y, x) \vee \neg X(s, y, x))))). \tag{5}$$

*Completeness.* For any two alternatives in any profile, all preferences of individuals and society must satisfy completeness:

$$\forall X(\boldsymbol{P}(X) \rightarrow \forall w(H(w) \rightarrow \forall x \forall y((A(x) \wedge A(y)) \rightarrow (X(w,x,y) \vee X(w,y,x))))),\tag{6}$$

$$\forall X(\boldsymbol{P}(X) \rightarrow \forall x \forall y((A(x) \wedge A(y)) \rightarrow (X(s,x,y) \vee X(s,y,x)))),\tag{7}$$

where (6) states the completeness of individuals, and (7) states that of society.

*Transitivity.* For any three alternatives in any profile, all preferences of individuals and society must satisfy transitivity:

$$\forall X(\boldsymbol{P}(X) \rightarrow \forall w(H(w) \rightarrow \forall x \forall y \forall z((A(x) \wedge A(y) \wedge A(z)) \\ \rightarrow ((X(w,x,y) \wedge X(w,y,z)) \rightarrow X(w,x,z)))))),\tag{8}$$

$$\forall X(\boldsymbol{P}(X) \rightarrow \forall x \forall y \forall z((A(x) \wedge A(y) \wedge A(z)) \\ \rightarrow ((X(s,x,y) \wedge X(s,y,z)) \rightarrow X(s,x,z)))).\tag{9}$$

*Unanimity* (*Pareto Property*). For any two alternatives in any profile, alternative $\alpha$ is strictly preferred to alternative $\beta$ in society if all individuals strictly prefer $\alpha$ to $\beta$:

$$\forall X(\boldsymbol{P}(X) \rightarrow \forall x \forall y((A(x) \wedge A(y)) \\ \rightarrow (\forall w(H(w) \rightarrow (X(w,x,y) \wedge \neg X(w,y,x))) \rightarrow (X(s,x,y) \wedge \neg X(s,y,x))))).\tag{10}$$

*Independence of Irrelevant Alternatives (IIA)*. If every individual keeps their pairwise preference relation unchanged between two or more profiles, the social preference over the two alternatives remains the same between these profiles:

$$\forall X \forall Y((\boldsymbol{P}(X) \wedge \boldsymbol{P}(Y)) \rightarrow \forall x \forall y((A(x) \wedge A(y)) \\ \rightarrow (\forall w(H(w) \rightarrow ((X(w,x,y) \leftrightarrow Y(w,x,y)) \wedge (X(w,y,x) \leftrightarrow Y(w,y,x)))) \\ \rightarrow ((X(s,x,y) \leftrightarrow Y(s,x,y)) \wedge (X(s,y,x) \leftrightarrow Y(s,y,x)))))).\tag{11}$$

*Non-dictatorship*. A dictator is a unique individual whose strict preference over two alternatives prevails as the social preference for any pair of alternatives in any profile. The statement that no dictator exists is translated as:

$$\neg \exists w(H(w) \\ \wedge \forall X(\boldsymbol{P}(X) \\ \rightarrow \forall x \forall y((A(x) \wedge A(y)) \rightarrow ((X(w,x,y) \wedge \neg X(w,y,x)) \rightarrow (X(s,x,y) \wedge \neg X(s,y,x))))) \\ \wedge \forall u(H(u) \rightarrow (\forall X(\boldsymbol{P}(X) \rightarrow \forall x \forall y((A(x) \wedge A(y)) \\ \rightarrow ((X(u,x,y) \wedge \neg X(u,y,x)) \rightarrow (X(s,x,y) \wedge \neg X(s,y,x)))))) \rightarrow u=w))).\tag{12}$$

Subformula $H(w)$ states that the entity is an individual. $\forall X(\ldots \neg X(s,y,x)))))$ states that for any pair of alternatives in any profile, that entity's strict preference constitutes the social preference. $\forall u(\ldots u=w))$ states that only one such entity exists. The statement that a dictator exists is denoted by $\neg$(12) thereafter. We should note that the formula that lacks $\forall u(\ldots u=w))$, the last subformula, in $\neg$(12) states that one or more individuals exist, each of whose strict preference coincides with a corresponding social preference; it does not represent the statement of dictatorship by a single person.

The theorem argues that no social welfare function exists under (1)–(12) but removing (12) allows social welfare functions characterized by dictatorship—and no others.

## 3.2. An illustration of proof procedure in logic

This subsection presents a simple example of a proof calculus to illustrate the basic idea. Let us consider the following statement: all ancient Greek philosophers are great, and Aristotle is an ancient Greek philosopher; thus, Aristotle is great. This inference is intuitively clear. The argument is structured as follows:

$$\forall x(P(x) \rightarrow G(x))$$
$$P(a) \qquad /G(a)$$

where $P(x)$ states that $x$ is an ancient Greek philosopher, $G(x)$ states that $x$ is great, and $P(a)$ says that Aristotle, denoted as $a$, is an ancient Greek philosopher. A proof calculus derives a conclusion, $G(a)$, from premises, $\{\forall x(P(x) \rightarrow G(x)), P(a)\}$, using axioms and inference rules adopted by the calculus. We use a proof calculus called natural deduction in this study, and this system has no axioms; the conclusion is derived only by applying inference rules to premises. The present example requires two rules, Universal Instantiation (UI) and Modus Ponens (MP). UI's idea is that since $P(x) \rightarrow G(x)$ holds for any $x$, the formula continues to hold under any instantiation of $x$. In this case, Aristotle is substituted into the formula; the universal quantifier is removed, and $P(a) \rightarrow G(a)$ is derived. MP is our natural daily inference that, as A⇒B and A hold, B follows. Since we now have $\{P(a) \rightarrow G(a), P(a)\}, \rightarrow$ can be eliminated by using MP, deriving $G(a)$. The proof diagram of this derivation is described as:

1 | $\forall x(P(x) \rightarrow G(x)$ prem.

2 | $P(a)$ prem.

3 | $P(a) \rightarrow G(a)$ 1, ($\forall E$)

4 | $G(a)$ 2, 3, ($\rightarrow E$)

 The form of the diagram described above is a version of so-called Jaśkowski-Fitch style. Starting from premises at the top, inference rules are applied sequentially to arrive at a conclusion. When a conclusion is deducible from premises, we say that the set of premises entails the conclusion. The entailment in the present example is written as $\{\forall x(P(x) \rightarrow G(x), P(a)\} \vdash G(a)$ in the symbolized form. In section 4, we aim to construct proof diagrams in which premises, reformulated according to the specified numbers of individuals and alternatives, entail dictatorship, ¬(12). The Greek-philosopher argument is constructed based on first-order logic, and only two inference rules are used. For the proof of the impossibility theorem, operations at the second-order level and rules other than the two are required. However, the proof system of natural deduction for second-order logic is constructed by a simple extension of the one for first-order logic [48]. The deductive procedure developed in the proof calculus ensures that every profile of all possible social welfare functions is examined, thereby establishing the theorem. As explained in the first section, the soundness of second-order logic guarantees that a provable inference is also semantically valid, which means that ¬(12) is always true when the premises remain true; no counterexample exists in which the premises are true and dictatorship is false. Thus, the correctness of inference as intuitively conceived, which closely aligns with semantic validity, is also confirmed.

## 4. Proof of the theorem

We first present a proof for the case of two individuals and three alternatives ($I = 2$ and $J = 3$). Then, the case of $I = 3$ and $J = 3$ is discussed. Finally, we reveal that the theorem holds in any case of $I > 2$ and $J > 3$. As we shall see, the simple extensions of the proof diagram for the case of $I = 2$ and $J = 3$ produce diagrams for cases assuming more individuals and alternatives. Each step begins with a non-technical overview, followed by a technical explanation of the corresponding proof diagram. The subsections on technical details may be skipped if one is concerned with the broad outline of the proof.

### 4.1. Two individuals and three alternatives ($I = 2$ and $J = 3$).

**4.1.1. Preliminaries.** We now consider the case of two individuals and three alternatives. As mentioned in section 3.2, premises (1)–(4), (6), and (8) are replaced with formulas that specify individuals and alternatives; the names of two individuals—$p$ and $q$—and three alternatives—$a$, $b$, and $c$—are added to our language. (1) and (2) are instantiated into:

$$H(p) \wedge H(q) \wedge p \neq q \wedge \forall x(H(x) \rightarrow (x = p \vee x = q)), \tag{13}$$

$$A(a) \wedge A(b) \wedge A(c) \wedge a \neq b \wedge b \neq c \wedge c \neq a \wedge \forall x(A(x) \rightarrow (x = a \vee x = b \vee x = c)). \tag{14}$$

In the case of three alternatives, individual preference relations that satisfy unrestricted domain, completeness, and transitivity are straightforward: there are 13 possible preferences. For example, $p$'s preference $a \succ b \succ c$ in profile 1−1 is denoted as $(R_{1-1}(p, a, b) \wedge \neg R_{1-1}(p, b, a)) \wedge (R_{1-1}(p, b, c) \wedge \neg R_{1-1}(p, c, b)) \wedge (R_{1-1}(p, a, c) \wedge \neg R_{1-1}(p, c, a)) \wedge R_{1-1}(p, a, a) \wedge R_{1-1}(p, b, b) \wedge R_{1-1}(p, c, c)$; subformula $R_{1-1}(p, a, a) \wedge R_{1-1}(p, b, b) \wedge R_{1-1}(p, c, c)$ represents reflexivity. If $q$ has the same preference, the profile is written as:

$(R_{1-1}(p, a, b) \wedge \neg R_{1-1}(p, b, a)) \wedge (R_{1-1}(p, b, c) \wedge \neg R_{1-1}(p, c, b)) \wedge (R_{1-1}(p, a, c) \wedge \neg R_{1-1}(p, c, a)) \wedge R_{1-1}(p, a, a) \wedge R_{1-1}(p, b, b) \wedge R_{1-1}(p, c, c) \wedge (R_{1-1}(q, a, b) \wedge \neg R_{1-1}(q, b, a)) \wedge (R_{1-1}(q, b, c) \wedge \neg R_{1-1}(q, c, b)) \wedge (R_{1-1}(q, a, c) \wedge \neg R_{1-1}(q, c, a)) \wedge R_{1-1}(q, a, a) \wedge R_{1-1}(q, b, b) \wedge R_{1-1}(q, c, c)$.

Since an individual has 13 possible preferences, the total number of profiles is 169 for two individuals and three alternatives. Then, (4), (6), and (8) are replaced by 169 formulas in a similar manner to profile 1−1, each specifying a profile. Finally, (3) is instantiated as:

$$P(R_{1-1}) \wedge \ldots \wedge P(R_{13-13}) \wedge R_{1-1} \neq R_{1-2} \wedge \cdots \wedge R_{13-12} \neq R_{13-13}$$
$$\wedge \forall X(P(X) \rightarrow (X = R_{1-1} \vee \ldots \vee X = R_{13-13})). \tag{15}$$

(5), (7), (9)–(11), (13)–(15), and 169 formulas specifying profiles are the premises of the deduction. $\Gamma$ denotes the set of these premises.

**The Impossibility Theorem ($I = 2$ and $J = 3$).** *In a society in which two individuals exist and have three alternatives, any social welfare function that satisfies the unrestricted domain, completeness, transitivity, unanimity, and IIA is dictatorial.*

**Proof.** A derivation to prove the sequent $\Gamma \vdash \neg(12)$ is described in <u>S1 Appendix</u>. Q.E.D.

**4.1.2. Non-technical overview.** Our strategy for deriving $\neg(12)$ from $\Gamma$ begins by assuming non-dictatorship (12) and selecting a profile. We then reveal that any possible case of social preference in that profile leads to contradiction under premises $\Gamma$ and assumption (12). Finally, it is shown that rejecting (12) resolves contradiction, but dictatorship is inevitable under $\Gamma$; hence, $\Gamma$ entails $\neg(12)$.

Specifically, the present proof selects the profile in which $p$'s preference is $\{a \succ b, b \succ c, a \succ c\}$ and $q$'s preference is $\{a \succ b, b \prec c, a \succ c\}$, and we focus on $b$ and $c$. All logically possible social preferences for the pair are $b \sim c$, $b \succ c$, $b \prec c$, and non-comparable one, which is denoted by $b \parallel c$: $R(s, b, c) \wedge R(s, c, b)$, $R(s, b, c) \wedge \neg R(s, c, b)$, $\neg R(s, b, c) \wedge R(s, c, b)$, and $\neg R(s, b, c) \wedge \neg R(s, c, b)$. The four cases are examined respectively, and a contradiction is derived in every case. Case $b \sim c$ violates transitivity. Cases $b \succ c$ and $b \prec c$ only yield social welfare functions characterized by dictatorship, thus violating the initial assumption (12). $b \parallel c$ trivially violates the completeness of social preference. Thus, $\Gamma$ causes a contradiction in the first and fourth cases, while (12) does so in the second and third cases. Removing either $\Gamma$ (or part of $\Gamma$) or (12) resolves contradictions and makes social welfare functions possible, but our choice of rejecting (12) while maintaining $\Gamma$ resolves the contradictions arising in cases $b \succ c$ and $b \prec c$, allowing dictatorships alone. Hence, dictatorship is concluded under $\Gamma$: $\Gamma \vdash \neg(12)$. Using a quasi-diagram, the flow of the proof is described as follows:

| premises Γ

|  | the assumption of non-dictatorship: (12)

|  |  | case $b \sim c$: the violation of transitivity

|  |  | case $b \succ c$: the violation of (12) ($p$'s dictatorships)

|  |  | case $b \prec c$: the violation of (12) ($q$'s dictatorships)

|  |  | case $b \parallel c$: the violation of completeness

|  | all possible cases produce contradiction under Γ and (12).

| ¬(12): dictatorship is concluded under premises Γ by rejecting (12).

**4.1.3. Technical explanation.** The following diagram provides a summary of the derivation in the proof described in S1 Appendix (the line numbers in the diagram correspond to those of the proof in the appendix):

1–177 | Γ̲̲̲̲̲̲̲̲̲ prem.

178 |  | (12)̲̲̲̲̲̲̲̲ prem.

361 |  | $R_{1-2}(s, b, c) \vee \neg R_{1-2}(s, b, c)$

362 |  |  | R̲1̲-̲2̲(̲s̲,̲ ̲b̲,̲ ̲c̲) prem.

363 |  |  | $R_{1-2}(s, c, b) \vee \neg R_{1-2}(s, c, b)$

364 |  |  |  | R̲1̲-̲2̲(̲s̲,̲ ̲c̲,̲ ̲b̲) prem.

462 |  |  |  | ⊥ (violating the transitivity of a social preference)

463 |  |  | $R_{1-2}(s, c, b) \rightarrow \perp$

464 |  |  |  | ¬̲R̲1̲-̲2̲(̲s̲,̲ ̲c̲,̲ ̲b̲) prem.

705 |  |  |  | $p$'s non-dictatorship

1490 |  |  |  | ⊥ (violating $p$'s non-dictatorship; $p$ is a dictator)

1491 |  |  | $\neg R_{1-2}(s, c, b) \rightarrow \perp$

1492 |  |  | ⊥

1493 |  | $R_{1-2}(s, b, c) \rightarrow \perp$

1494 |  |  | ¬̲R̲1̲-̲2̲(̲s̲,̲ ̲b̲,̲ ̲c̲) prem.

1495 |  |  | $R_{1-2}(s, c, b) \vee \neg R_{1-2}(s, c, b)$

1496 |  |  |  | R̲1̲-̲2̲(̲s̲,̲ ̲c̲,̲ ̲b̲) prem.

2293 |  |  |  | ⊥ (violating $q$'s non-dictatorship; $q$ is a dictator)

2294 |  |  | $R_{1-2}(s, c, b) \rightarrow \perp$

2295 |  |  |  | ¬̲R̲1̲-̲2̲(̲s̲,̲ ̲c̲,̲ ̲b̲) prem.

2308 |  |  |  | ⊥(violating the completeness of a social preference)

2309 |   |   | $\neg R_{1-2}(s, c, b) \rightarrow \perp$

2310 |   |   | $\perp$

2311 |   | $\neg R_{1-2}(s, b, c) \rightarrow \perp$

2312 |   | $\perp$

2313 | $\neg(12)$

Lines 1–177 are the premises of the argument: $\Gamma$. Under these premises, the non-existence of a dictator is assumed in line 178. The deduction first chooses a profile in which an individual strictly prefers an alternative to another while another individual has the opposite preference. In this deduction, profile $R_{1-2}$ in line 5 is chosen, and alternatives $b$ and $c$ are used for two such alternatives; in $R_{1-2}$, individual $p$'s preference over $b$ and $c$ is $R_{1-2}(p, b, c) \wedge \neg R_{1-2}(p, c, b)$, whereas $q$'s preference is $\neg R_{1-2}(q, b, c) \wedge R_{1-2}(q, c, b)$.

The number of (truly) logically possible social preferences over $b$ and $c$ is four: $R_{1-2}(s, b, c) \wedge R_{1-2}(s, c, b)$, $R_{1-2}(s, b, c) \wedge \neg R_{1-2}(s, c, b)$, $\neg R_{1-2}(s, b, c) \wedge R_{1-2}(s, c, b)$, and $\neg R_{1-2}(s, b, c) \wedge \neg R_{1-2}(s, c, b)$. The four cases are successively examined in the deduction. On the assumption of $R_{1-2}(s, b, c)$ in line 362, the social preferences might be either $R_{1-2}(s, c, b)$ or $\neg R_{1-2}(s, c, b)$, as stated in 363. Then, $R_{1-2}(s, b, c) \wedge R_{1-2}(s, c, b)$ and $R_{1-2}(s, b, c) \wedge \neg R_{1-2}(s, c, b)$ are examined in lines 364–463 and 464–1491, respectively. Similarly, assuming $\neg R_{1-2}(s, b, c)$ in line 1494, $\neg R_{1-2}(s, b, c) \wedge R_{1-2}(s, c, b)$ and $\neg R_{1-2}(s, b, c) \wedge \neg R_{1-2}(s, c, b)$ are examined in lines 1496–2294 and 2295–2309, respectively.

In the first case, $R_{1-2}(s, b, c) \wedge R_{1-2}(s, c, b)$, the transitivity of social preference is violated. The violation in $R_{3-6}$ is derived in this deduction. Thus, the assumption of $R_{1-2}(s, c, b)$ in line 364 produces a contradiction; $R_{1-2}(s, c, b) \rightarrow \perp$ is stated in line 463. In the second case, $R_{1-2}(s, b, c) \wedge \neg R_{1-2}(s, c, b)$, following the assumption stated in line 178 that no one is a dictator, line 705 instantiates $p$ as such a non-dictator. However, line 1490 states that the statement of $p$'s non-dictatorship produces a contradiction; under the assumption of $R_{1-2}(s, b, c) \wedge \neg R_{1-2}(s, c, b)$, $p$ is a dictator in every social welfare function that satisfies $\Gamma$. Then, $\neg R_{1-2}(s, c, b) \rightarrow \perp$ is stated in line 1491. Since both $R_{1-2}(s, b, c) \wedge R_{1-2}(s, c, b)$ and $R_{1-2}(s, b, c) \wedge \neg R_{1-2}(s, c, b)$ produce a contradiction, all cases of $R_{1-2}(s, b, c)$ yield a contradiction. Thus, assuming $R_{1-2}(s, b, c)$ in line 362 is a contradiction; $R_{1-2}(s, b, c) \rightarrow \perp$ is stated in line 1493.

In the third case, $\neg R_{1-2}(s, b, c) \wedge R_{1-2}(s, c, b)$, since $p$ and $q$ are symmetrical, replacing $p$ with $q$ produces a contradiction similar to that in the second case. Thus, $R_{1-2}(s, c, b) \rightarrow \perp$ is stated in line 2294. Although the social preference's violation of completeness in the fourth case, $\neg R_{1-2}(s, b, c) \wedge \neg R_{1-2}(s, c, b)$, is trivial, lines 2295–2309 derive it formally; $\neg R_{1-2}(s, c, b) \rightarrow \perp$ is stated in line 2309. Since all cases of $\neg R_{1-2}(s, b, c)$ produce a contradiction, line 2311 states that $\neg R_{1-2}(s, b, c) \rightarrow \perp$.

Line 2312 states that any logically possible social preference in $R_{1-2}$ produces a contradiction under the assumption of non-dictatorship stated in line 178. Hence, dictatorship follows from $\Gamma$, as stated in line 2313; the theorem is established.

### 4.2. Three individuals and three alternatives ($I = 3$ and $J = 3$)

Next, we consider a society with three individuals having three alternatives.

**The Impossibility Theorem ($I$ = 3 and $J$ = 3).** *In a society in which three individuals exist and have three alternatives, any social welfare function that satisfies the unrestricted domain, completeness, transitivity, unanimity, and IIA is dictatorial.*

**4.2.1. Non-technical overview.** The proof proceeds in a similar manner to $I = 2$ and $J = 3$. The set of premises, $\Gamma'$, is expanded to accommodate the increased number of individuals and profiles. $r$ denotes the third individual. As in $I = 2$ and $J = 3$, we assume non-dictatorship (12) at the top and select a profile in which $p$'s preference is $\{a \succ b, b \succ c, a \succ c\}$ while the preferences of the other two are $\{a \succ b, b \prec c, a \succ c\}$. We call it profile P. The four cases of social preference in profile P, each prefixed with case (P), are examined. Case (P) $b \sim c$ violates transitivity, case (P) $b \parallel c$ violates completeness, and case (P) $b \succ c$ violates (12). Then, under case (P) $b \prec c$, we move to profile Q with $p$ and $q$ interchanged. In profile

Q, cases (Q) $b \sim c$ and $b \parallel c$ violate transitivity and completeness, respectively, and case (Q) $b \succ c$ violates (12). Finally, profile R with $q$ and $r$ swapped is examined under case (P) $b \prec c$ and case (Q) $b \prec c$. Cases (R) $b \sim c$ and $b \parallel c$ violate transitivity and completeness, respectively, while case (R) $b \succ c$ violates (12). Then, the only remaining case is (R) $b \prec c$ under case (P) $b \prec c$ and case (Q) $b \prec c$, in which no one is a dictator, but transitivity is violated. Thus, contradictions are derived in all possible cases. Removing either Γ′ or (12) makes social welfare functions possible, but the rejection of (12) while maintaining Γ′ resolves the contradictions that involve dictatorships only; hence, Γ′ entails ¬(12). The following quasi-diagram illustrates the nested structure constructed in cases $b \prec c$:

|Γ′ prem.

| | the assumption of non-dictatorship: (12)

| | | case (P) $b \sim c$: the violation of transitivity

| | | case (P) $b \succ c$: the violation of (12) ($p$'s dictatorship)

| | | case (P) $b \prec c$

| | | | case (Q) $b \sim c$: the violation of transitivity

| | | | case (Q) $b \succ c$: the violation of (12) ($q$'s dictatorship)

| | | | case (Q) $b \prec c$

| | | | | case (R) $b \sim c$: the violation of transitivity

| | | | | case (R) $b \succ c$: the violation of (12) ($r$'s dictatorship)

| | | | | case (R) $b \prec c$: the violation of transitivity

| | | | | case (R) $b \parallel c$: the violation of completeness

| | | | case (Q) $b \parallel c$: the violation of completeness

| | | case (P) $b \parallel c$: the violation of completeness

| | all possible cases produce contradiction under Γ′ and (12).

| ¬(12): dictatorship is concluded under premises Γ′ by rejecting (12).

**4.2.2. Technical explanation.** The name of the third individual, $r$, is added to our language. Let $R_k$ be the profile in which individual $k$ has $(R_k(k, a, b) \wedge \neg R_k(k, b, a)) \wedge (R_k(k, b, c) \wedge \neg R_k(k, c, b)) \wedge (R_k(k, a, c) \wedge \neg R_k(k, c, a))$, which is the same preference relation as $p$'s preference in $R_{1\text{-}2}$ of the proof for $I = 2$ and $J = 3$, while the rest of the individuals, denoted by $\neg k$, have $(R_k(\neg k, a, b) \wedge \neg R_k(\neg k, b, a)) \wedge (\neg R_k(\neg k, b, c) \wedge R_k(\neg k, c, b)) \wedge (R_k(\neg k, a, c) \wedge \neg R_k(\neg k, c, a))$, which is the same preference relation as $q$'s preference in $R_{1\text{-}2}$ of the proof for $I = 2$ and $J = 3$, where $k$ might be $p$, $q$, $r$, and reflexive relations are omitted. Γ′ denotes the set of premises that extends Γ to represent the case of three individuals by replacing 169 profiles with 2197 profiles, adding individual $r$ to (13), and reformulating (15) to have 2197 profiles.

The diagram below sketches a proof of the sequent Γ′ ⊢ ¬(12).

1 | Γ′ prem.

2 | | (12) prem.

3 | | $R_p(s, b, c) \vee \neg R_p(s, b, c)$

4 | | | $R_p(s, b, c)$ prem.

5 | | | $R_p(s, c, b) \lor \neg R_p(s, c, b)$

6 | | | | <u>$R_p(s, c, b)$ prem.</u>

7 | | | $R_p(s, c, b) \to \bot$ (the violation of transitivity)

8 | | | | <u>$\neg R_p(s, c, b)$ prem.</u>

9 | | | $\neg R_p(s, c, b) \to \bot$ ($p$'s dictatorship)

10 | | | $\bot$

11 | | $R_p(s, b, c) \to \bot$

12 | | | <u>$\neg R_p(s, b, c)$ prem.</u>

13 | | | $R_p(s, c, b) \lor \neg R_p(s, c, b)$

14 | | | | <u>$R_p(s, c, b)$ prem.</u>

15 | | | $R_q(s, b, c) \lor \neg R_q(s, b, c)$

16 | | | | | <u>$R_q(s, b, c)$     prem.</u>

17 | | | | $R_q(s, c, b) \lor \neg R_q(s, c, b)$

18 | | | | | | <u>$R_q(s, c, b)$     prem.</u>

19 | | | | $R_q(s, c, b) \to \bot$ (the violation of transitivity)

20 | | | | | | <u>$\neg R_q(s, c, b)$ prem.</u>

21 | | | | $\neg R_q(s, c, b) \to \bot$ ($q$'s dictatorship)

22 | | | | | $\bot$

23 | | | $R_q(s, b, c) \to \bot$

24 | | | | | <u>$\neg Rq(s, b, c)$ prem.</u>

25 | | | | $R_q(s, c, b) \lor \neg R_q(s, c, b)$

26 | | | | | | <u>$R_q(s, c, b)$ prem.</u>

27 | | | | | $R_r(s, b, c) \lor \neg R_r(s, b, c)$

28 | | | | | | | <u>$R_r(s, b, c)$     prem.</u>

29 | | | | | | $R_r(s, c, b) \lor \neg R_r(s, c, b)$

30 | | | | | | | | <u>$R_r(s, c, b)$     prem.</u>

31 | | | | | | | $R_r(s, c, b) \to \bot$ (the violation of transitivity)

32 | | | | | | | | <u>$\neg R_r(s, c, b)$ prem.</u>

33 | | | | | | | $\neg R_r(s, c, b) \to \bot$ ($r$'s dictatorship)

34 | | | | | | | | $\bot$

35 | | | | | | $R_r(s, b, c) \to \bot$

36 | | | | | | | | $\neg R_{t}(s, b, c)$ prem.

37 | | | | | | | | $R_{r}(s, c, b) \lor \neg R_{r}(s, c, b)$

38 | | | | | | | | | $R_{t}(s, c, b)$ prem.

39 | | | | | | | | $R_{r}(s, c, b) \to \bot$ (the violation of transitivity)

40 | | | | | | | | | $\neg R_{t}(s, c, b)$ prem.

41 | | | | | | | | $\neg R_{r}(s, c, b) \to \bot$ (the violation of completeness)

42 | | | | | | | | $\bot$

43 | | | | | | | $\neg R_{r}(s, b, c) \to \bot$

44 | | | | | | | $\bot$

45 | | | | | $R_{q}(s, c, b) \to \bot$

46 | | | | | | $\neg R_{q}(s, c, b)$ prem.

47 | | | | | $\neg R_{q}(s, c, b) \to \bot$ (the violation of completeness)

48 | | | | | $\bot$

49 | | | | $\neg R_{q}(s, b, c) \to \bot$

50 | | | | $\bot$

51 | | | $R_{p}(s, c, b) \to \bot$

52 | | | | $\neg R_{p}(s, c, b)$ prem.

53 | | | $\neg R_{p}(s, c, b) \to \bot$ (the violation of completeness)

54 | | | $\bot$

55 | | $\neg R_{p}(s, b, c) \to \bot$

56 | | $\bot$

57 | $\neg (12)$

The proof for $I = 3$ and $J = 3$ can be constructed with the simple extension of the proof for $I = 2$ and $J = 3$, except that it is nested in the cases of $\neg R_{k}(s, b, c) \land R_{k}(s, c, b)$. As stated in line 3, the derivation begins with $R_{p}(s, b, c) \lor \neg R_{p}(s, b, c)$, which corresponds to $R_{1\text{-}2}(s, b, c) \lor \neg R_{1\text{-}2}(s, b, c)$ in line 361 of the proof for $I = 2$ and $J = 3$. As with $R_{1\text{-}2}$, $R_{p}$ has four logically possible social preferences over $b$ and $c$: $R_{p}(s, b, c) \land R_{p}(s, c, b)$, $R_{p}(s, b, c) \land \neg R_{p}(s, c, b)$, $\neg R_{p}(s, b, c) \land R_{p}(s, c, b)$, and $\neg R_{p}(s, b, c) \land \neg R_{p}(s, c, b)$. The four cases are successively examined in a manner similar to the proof for $I = 2$ and $J = 3$.

**$R_{p}(s, b, c) \land R_{p}(s, c, b)$**: Any function that has three individuals includes the profiles in which an individual has the same preference relation as $p$'s in the two-individual case and the rest of the individuals have the same preference relation as $q$'s in the case. Since the corresponding profiles in the two-individual case produce the violation of transitivity in social preference, any function having three individuals also does so, as stated in line 7.

**$R_{p}(s, b, c) \land \neg R_{p}(s, c, b)$**: The proof for $I = 2$ and $J = 3$ reveals that once $R_{1\text{-}2}(s, b, c) \land \neg R_{1\text{-}2}(s, c, b)$ is assumed, every $p$'s strict preference coincides with the social preference. To illustrate this process, consider profiles $R_{4\text{-}5}$, $R_{4\text{-}6}$, and $R_{4\text{-}8}$ in the two-individual case, where $p$ has $(\neg R(p, a, b) \land R(p, b, a)) \land (R(p, b, c) \land \neg R(p, c, b)) \land (\neg R(p, a, c) \land R(p, c, a))$ and $q$ has

$(\neg R(q, b, c) \wedge R(q, c, b)) \wedge (\neg R(q, a, c) \wedge R(q, c, a))$. $q$'s preferences over $a$ and $b$ are $R_{4\text{-}5}(q, a, b) \wedge \neg R_{4\text{-}5}(q, b, a)$, $\neg R_{4\text{-}6}(q, a, b) \wedge R_{4\text{-}6}(q, b, a)$, and $R_{4\text{-}8}(q, a, b) \wedge R_{4\text{-}8}(q, b, a)$, respectively.

Since $(R_{1\text{-}2}(p, b, c) \wedge \neg R_{1\text{-}2}(p, c, b)) \wedge (\neg R_{1\text{-}2}(q, b, c) \wedge R_{1\text{-}2}(q, c, b))$, IIA diffuses $R_{1\text{-}2}(s, b, c) \wedge \neg R_{1\text{-}2}(s, c, b)$ to the three profiles and determines $R_{4\text{-}5}(s, b, c) \wedge \neg R_{4\text{-}5}(s, c, b)$, $R_{4\text{-}6}(s, b, c) \wedge \neg R_{4\text{-}6}(s, c, b)$, and $R_{4\text{-}8}(s, b, c) \wedge \neg R_{4\text{-}8}(s, c, b)$. Unanimity determines $\neg R_{4\text{-}5}(s, a, c) \wedge R_{4\text{-}5}(s, c, a)$, $\neg R_{4\text{-}6}(s, a, c) \wedge R_{4\text{-}6}(s, c, a)$, and $\neg R_{4\text{-}8}(s, a, c) \wedge R_{4\text{-}8}(s, c, a)$. Then, transitivity determines $\neg R_{4\text{-}5}(s, a, b) \wedge R_{4\text{-}5}(s, b, a)$, $\neg R_{4\text{-}6}(s, a, b) \wedge R_{4\text{-}6}(s, b, a)$, and $\neg R_{4\text{-}8}(s, a, b) \wedge R_{4\text{-}8}(s, b, a)$. $(\neg R_{4\text{-}6}(s, a, b) \wedge R_{4\text{-}6}(s, b, a)$ can also be determined by unanimity. The deduction in S1 Appendix uses unanimity, as stated in line 257.) Using IIA, those determined by transitivity are diffused to the social preferences, each of whose profiles over $a$ and $b$ is either $((\neg R(p, a, b) \wedge R(p, b, a)) \wedge (R(q, a, b) \wedge \neg R(q, b, a)))$, $((\neg R(p, a, b) \wedge R(p, b, a)) \wedge (\neg R(q, a, b) \wedge R(q, b, a)))$, or $((\neg R(p, a, b) \wedge R(p, b, a)) \wedge (R(q, a, b) \wedge R(q, b, a)))$. Thereafter, in some of the profiles to which $\neg R(s, a, b) \wedge R(s, b, a)$ has been assigned, the other social preferences are similarly determined by unanimity and transitivity. Again, social preferences determined by transitivity are diffused to other profiles by IIA. Repeating similar steps eventually derives $p$'s dictatorship.

We should note that since the social preferences diffused by IIA are determined by the transitivity of social preference (except for the initial assumption, $R_{1\text{-}2}(s, b, c) \wedge \neg R_{1\text{-}2}(s, c, b)$), they do not depend on $q$'s individual preference over the two alternatives. In the above example, the social preferences over $a$ and $b$ that are assigned to $R_{4\text{-}5}$, $R_{4\text{-}6}$, and $R_{4\text{-}8}$ are the same irrespective of $q$'s preferences over $a$ and $b$ in $R_{4\text{-}5}$, $R_{4\text{-}6}$, and $R_{4\text{-}8}$.

We now consider the case of three individuals. Consider the profiles in which $p$ has $(\neg R(p, a, b) \wedge R(p, b, a)) \wedge (R(p, b, c) \wedge \neg R(p, c, b)) \wedge (\neg R(p, a, c) \wedge R(p, c, a))$, while the rest of the individuals have $\neg R(\neg p, b, c) \wedge R(\neg p, c, b)$ and $\neg R(\neg p, a, c) \wedge R(\neg p, c, a)$. Such profiles correspond to $R_{4\text{-}5}$, $R_{4\text{-}6}$, and $R_{4\text{-}8}$ in the two-individual case, but the number of profiles increases from three $(=3^1)$ to nine $(=3^2)$ due to the increase in the number of individuals. Like the above example, once $R_p(s, b, c) \wedge \neg R_p(s, c, b)$ is assumed, this social preference is diffused to the nine profiles by IIA. The social preferences over $a$ and $c$ in the nine profiles are determined to be $\neg R(s, a, c) \wedge R(s, c, a)$ by unanimity. Transitivity determines $\neg R(s, a, b) \wedge R(s, b, a)$ in the nine profiles irrespective of $q$ and $r$ individual preferences over $a$ and $b$. Then, using IIA, those determined by transitivity are diffused to the social preferences, each of whose profiles over $a$ and $b$ is any one of these nine profiles. Repeating similar steps eventually derives $p$'s dictatorship; it violates the non-dictatorship assumption in line 2, as stated in line 9.

$\neg R_p(s, b, c) \wedge R_p(s, c, b)$: $q$ and $r$ decide the social preference in this case, and they might be a dictator. Then, consider profile $R_q$ under the assumption of $\neg R_p(s, b, c) \wedge R_p(s, c, b)$, which starts from line 15.

$R_q(s, b, c) \wedge R_q(s, c, b)$: Similar to the case of $R_p(s, b, c) \wedge R_p(s, c, b)$, the violation of transitivity occurs, as stated in line 19.

$R_q(s, b, c) \wedge \neg R_q(s, c, b)$: Similar to the case of $R_p(s, b, c) \wedge \neg R_p(s, c, b)$, $q$'s dictatorship is established, as stated in line 21.

$\neg R_q(s, b, c) \wedge R_q(s, c, b)$: $r$ decides the social preferences over $b$ and $c$ in both $R_p$ and $R_q$; $r$ might be a dictator. Then, let us consider $R_r$ under the assumption of $\neg R_q(s, b, c) \wedge R_q(s, c, b)$, which starts from line 27.

$R_r(s, b, c) \wedge R_r(s, c, b)$: The violation of transitivity occurs, as stated in line 31.

$R_r(s, b, c) \wedge \neg R_r(s, c, b)$: $r$'s dictatorship is established, as stated in line 33.

$\neg R_r(s, b, c) \wedge R_r(s, c, b)$: No individual decides all three social preferences; no dictator exists. However, the transitivity of social preference is violated in these functions. To illustrate, consider profile $(\neg R_1(\neg r, a, b) \wedge R_1(\neg r, b, a)) \wedge (\neg R_1(\neg r, b, c) \wedge R_1(\neg r, c, b)) \wedge (\neg R_1(\neg r, a, c) \wedge R_1(\neg r, c, a)) \wedge (\neg R_1(r, a, b) \wedge R_1(r, b, a)) \wedge (R_1(r, b, c) \wedge \neg R_1(r, c, b)) \wedge (R_1(r, a, c) \wedge \neg R_1(r, c, a))$. Unanimity determines $\neg R_1(s, a, b) \wedge R_1(s, b, a)$. IIA diffuses $\neg R_r(s, b, c) \wedge R_r(s, c, b)$ to $R_1$. Transitivity determines $\neg R_1(s, a, c) \wedge R_1(s, c, a)$. Then, consider $(\neg R_2(\neg r, a, b) \wedge R_2(\neg r, b, a)) \wedge (R_2(\neg r, b, c) \wedge \neg R_2(\neg r, c, b)) \wedge (\neg R_2(\neg r, a, c) \wedge R_2(\neg r, c, a)) \wedge (R_2(r, a, b) \wedge \neg R_2(r, b, a)) \wedge (R_2(r, b, c) \wedge \neg R_2(r, c, b)) \wedge (R_2(r, a, c) \wedge \neg R_2(r, c, a))$. Unanimity determines $R_2(s, b, c) \wedge \neg R_2(s, c, b)$. IIA diffuses $\neg R_1(s, a, c) \wedge R_1(s, c, a)$ to $R_2$. Transitivity determines $\neg R_2(s, a, b) \wedge R_1(s, b, a)$. For $(\neg R_3(p, a, b) \wedge R_3(p, b, a)) \wedge (\neg R_3(p, b, c) \wedge R_3(p, c, b)) \wedge (\neg R_3(p, a, c) \wedge R_3(p, c, a)) \wedge (\neg R_3(q, a, b) \wedge R_3(q, b, a)) \wedge (R_3(q, b, c) \wedge \neg R_3(q, c, b)) \wedge (R_3(q, a, c) \wedge \neg R_3(q, c, a)) \wedge (R_3(r, a, b) \wedge \neg R_3(r, b, a)) \wedge (\neg R_3(r, b, c) \wedge R_3(r, c, b)) \wedge (R_3(r, a, c) \wedge \neg R_1(r, c, a))$, IIA diffuses $\neg R_2(s, a, b) \wedge R_1(s, b, a)$ to $R_3$ while diffusing $\neg R_q(s, b, c) \wedge R_q(s, c, b)$ to $R_3$. Transitivity determines $\neg R_3(s, a, c) \wedge$

$R_3(s, c, a)$. For $(\neg R_4(p, a, b) \wedge R_4(p, b, a)) \wedge (R_4(p, b, c) \wedge \neg R_4(p, c, b)) \wedge (\neg R_4(p, a, c) \wedge R_4(p, c, a)) \wedge (R_4(\neg p, a, b) \wedge \neg R_4(\neg p, b, a)) \wedge (R_4(\neg p, b, c) \wedge \neg R_4(\neg p, c, b)) \wedge (R_4(\neg p, a, c) \wedge \neg R_4(\neg p, c, a))$, unanimity determines $R_4(s, b, c) \wedge \neg R_4(s, c, b)$. IIA diffuses $\neg R_3(s, a, c) \wedge R_3(s, c, a)$ to $R_4$. Transitivity determines $\neg R_4(s, a, b) \wedge R_4(s, b, a)$. Then, consider $(\neg R_5(p, a, b) \wedge R_5(p, b, a)) \wedge (R_5(p, b, c) \wedge \neg R_5(p, c, b)) \wedge (R_5(p, a, c) \wedge \neg R_5(p, c, a)) \wedge (\neg R_5(\neg p, a, b) \wedge \neg R_5(\neg p, b, a)) \wedge (\neg R_5(\neg p, b, c) \wedge R_5(\neg p, c, b)) \wedge (R_5(\neg p, a, c) \wedge \neg R_5(\neg p, c, a))$. Unanimity determines $R_5(s, a, c) \wedge \neg R_5(s, c, a)$. IIA diffuses $\neg R_p(s, b, c) \wedge R_p(s, c, b)$ to $R_5$. Transitivity determines $R_5(s, a, b) \wedge \neg R_5(s, b, a)$. However, IIA also diffuses $\neg R_4(s, a, b) \wedge R_4(s, b, a)$ to $R_5$; $\neg R_5(s, a, b) \wedge R_5(s, b, a)$ violates transitivity, as stated in line 39.

$\neg R_r(s, b, c) \wedge \neg R_r(s, c, b)$: Completeness is violated, as stated in line 41.

Thus, all possible $R_r$'s social preferences produce contradictions if $\neg R_q(s, b, c) \wedge R_q(s, c, b)$ is assumed.

Hence, this assumption yields the contradictions in the first place, as stated in line 45.

$\neg R_q(s, b, c) \wedge \neg R_q(s, c, b)$: Completeness is violated, as stated in line 47.

Then, all possible $R_q$'s social preferences produce contradictions if $\neg R_p(s, b, c) \wedge R_p(s, c, b)$ is assumed.

Hence, this assumption yields the contradictions in the first place, as stated in line 51.

$\neg R_p(s, b, c) \wedge \neg R_p(s, c, b)$: Completeness is violated, as stated in line 53.

All possible $R_p$'s social preferences produce contradictions under the non-dictatorship assumption in line 2. Then, this assumption yields the contradictions in the first place. Hence $\Gamma'$ entails $\neg(12)$: $\Gamma' \vdash \neg(12)$, as stated in line 57.

### 4.3. The full impossibility theorem

Finally, we consider the full version of the theorem.

*The Impossibility Theorem (I ≥ 2 and J ≥ 3)* *In a society in which two or more individuals exist and they have three or more alternatives, any social welfare function that satisfies the unrestricted domain, completeness, transitivity, unanimity, and IIA is dictatorial.*

**4.3.1. Non-technical overview.** The set of premises, $\Gamma''$, expands with the increasing number of individuals and alternatives. The nested structure constructed in cases $b \prec c$ of the proof for $I = 3$ and $J = 3$ becomes deeper with more individuals, and the derivation becomes longer with more alternatives. However, the additional procedures simply replicate components of the earlier proofs, and $\neg(12)$ is concluded.

| | premises $\Gamma''$

| | the assumption of non-dictatorship: (12)

| | | profile P

| | | | profile Q

| | | | | profile R

⋮

| … | profile I

⋮

| | | | | profile R

| | | | profile Q

| | | profile P

| | all possible cases produce contradiction under $\Gamma''$ and (12).

| ¬(12): dictatorship is concluded under premises $\Gamma''$ by rejecting (12).

 

**4.3.2. Technical explanation.** We discuss the remaining cases: $I > 3$ and $J > 3$. In the diagram, $\Gamma''$ denotes the set of premises for $I$ individuals and $J$ alternatives, which is formulated in a manner similar to $\Gamma'$. In a society that has more than three individuals, more profiles must be examined, and the derivation in $\neg R_k(s, b, c) \wedge R_k(s, c, b)$ is nested more deeply; vertical and horizontal ellipses in the diagram below represent such nests. However, since all individuals have the same quality, the same deductive procedure as that of the proof for $I = 3$ and $J = 3$ unfolds irrespective of the number of individuals. Regarding the number of alternatives, since preference relations between alternatives comprise pairwise relations among three alternatives, any preference relations that include more than three alternatives are decomposed into triples; the argument on three alternatives is maintained in any subsets of three alternatives taken from $J$ alternatives. Thus, although a longer derivation is required for a greater number of alternatives, the procedure similar to the three-alternative case holds for any number of alternatives greater than three. Hence, the impossibility theorem for any case of $I > 3$ and $J > 3$ is established by constructing a nested diagram as displayed below.

1 | $\underline{\Gamma''}$ ________ prem.

2 | | $\underline{(12)}$ ________ prem.

3 | | $R_p(s, b, c) \vee \neg R_p(s, b, c)$

4 | | | $\underline{R_p(s, b, c)}$ ________ prem.

5 | | | $R_p(s, c, b) \rightarrow \bot$(the violation of transitivity)

6 | | | $\neg R_p(s, c, b) \rightarrow \bot$ ($p$'s dictatorship)

7 | | | $\bot$

8 | | $R_p(s, b, c) \rightarrow \bot$

9 | | | $\underline{\neg R_p(s, b, c) \text{ prem.}}$

10 | | | | $\underline{R_p(s, c, b) \text{ prem.}}$

11 | | | | $R_q(s, b, c) \vee \neg R_q(s, b, c)$

12 | | | | | $\underline{R_q(s, b, c)}$ ________ prem.

13 | | | | | $R_q(s, c, b) \rightarrow \bot$(the violation of transitivity)

14 | | | | | $\neg R_q(s, c, b) \rightarrow \bot$($q$'s dictatorship)

15 | | | | | $\bot$

16 | | | | $R_q(s, b, c) \rightarrow \bot$

17 | | | | | $\underline{\neg R_q(s, b, c) \text{ prem.}}$

18 | | | | | | $\underline{R_q(s, c, b) \text{ prem.}}$

 $\vdots$

19 | ... | $R_i(s, b, c) \vee \neg R_i(s, b, c)$

20 | ... | | $\underline{R_i(s, b, c)}$ ___ prem.

21 | ... | | $R_i(s, c, b) \rightarrow \bot$ (the violation of transitivity)

22 | ... | | $\neg R_i(s, c, b) \rightarrow \bot$ ($I$'s dictatorship)

23 |                    …                   |   |⊥

24 |                    …                   | $R_i(s, b, c) \to \bot$

25 |                    …                   |   | ¬$R_i(s, b, c)$ prem.

26 |                    …                   |   | $R_i(s, c, b) \to \bot$ (the violation of transitivity)

27 |                    …                   |   | ¬$R_i(s, c, b) \to \bot$ (the violation of completeness)

28 |                    …                   |   |⊥

29 |                    …                   | ¬$R_i(s, b, c) \to \bot$

30 |                    …                   | ⊥

⋮

31 |  |  |  |   | $R_q(s, c, b) \to \bot$

32 |  |  |  |   | ¬$R_q(s, c, b) \to \bot$ (the violation of completeness)

33 |  |  |  |   |⊥

34 |  |  |   | ¬$R_q(s, b, c) \to \bot$

35 |  |  |   |⊥

36 |  |   | $R_p(s, c, b) \to \bot$

37 |  |   | ¬$R_p(s, c, b) \to \bot$ (the violation of completeness)

38 |  |   |⊥

39 |   | ¬$R_p(s, b, c) \to \bot$

40 |   |⊥

41 | ¬(12)

## 5. Conclusion

Previous studies have sought to enhance the comprehension of the theorem's proof by devising ideas such as decisive sets and pivotal voters. While the approaches employed differ, the present study aligns with the earlier ones in pursuing the same purpose. Decisive sets and pivotal voters significantly facilitate intuitive insight into the theorem. Meanwhile, the proof developed using formal logic proceeds only with formulas, eliminating verbal ambiguity. Moreover, the whole composition of the proof comprises case analysis with a simple nest, facilitating a better grasp of the global structure of the social welfare function. Thus, both contribute to studies on the theorem in effective ways.

Microeconomics usually assumes completeness in preference relations, thereby focusing on agents whose preferences are defined by ∼, ≻, and ≺. However, economists dissatisfied with this standard assumption have investigated preference orders that include broader relations [49,50]. We also explicitly consider such a relation: non-comparability, denoted as ||. As shown in the proof, logic techniques effectively preserve all logically possible patterns, not limited to those assumed rational in economics. Thus, the approach employed in this study interacts synergistically with economists' efforts to have a broader perspective on economic agents.

Finally, analytic philosophy has traditionally made use of formal logic. At the same time, mathematical logicians have applied their tools to philosophical problems, and a research area known as philosophical logic has been established.

Thus, philosophy and logic have been closely connected, but their studies primarily focused on epistemological issues about mind, language, and natural sciences. By extending the applicability of mathematical logic to disciplines addressing social mechanisms, the type of research presented herein would help promote integrative inquiry across philosophy, the social sciences, and mathematics.

## Supporting information

**S1 Appendix. A proof of the impossibility theorem (two individuals and three alternatives).**
(PDF)

## Acknowledgments

I am grateful to Shigeru Fujimoto for supporting my studies over the years.

## Author contributions

**Conceptualization:** Kazuya Yamamoto.

**Formal analysis:** Kazuya Yamamoto.

**Investigation:** Kazuya Yamamoto.

**Methodology:** Kazuya Yamamoto.

**Project administration:** Kazuya Yamamoto.

**Resources:** Kazuya Yamamoto.

**Supervision:** Kazuya Yamamoto.

**Validation:** Kazuya Yamamoto.

**Writing – original draft:** Kazuya Yamamoto.

**Writing – review & editing:** Kazuya Yamamoto.

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
