## [Decision Letter · Decision Letter 0]

22 Aug 2025

Dear Dr. Yamamoto,

Thank you for submitting your manuscript to PLOS ONE. After careful consideration, we feel that it has merit but does not fully meet PLOS ONE’s publication criteria as it currently stands. Therefore, we invite you to submit a revised version of the manuscript that addresses the points raised during the review process.

We look forward to receiving your revised manuscript.

Kind regards,

Yu kun Wang

Academic Editor

PLOS ONE

Journal Requirements:

**Additional Editor Comments:**

This paper is technically sound and has strong, albeit somewhat limited in application, conclusions. It certainly merits publication, although it needs some reworking. Please consider the following:

0 – General:

Although the language you employ is mostly precise, the text is very dry. I understand your approach of conducting a full logical proof with minimal use of verbal remarks, but it can alienate non-specialist readers. This level of abstraction works against the broader dissemination of your work. Please consider adding illustrative examples and descriptions less reliant on pure logical expressions.

1 – Abstract

Your abstract is very short. It fails to properly place your work within the existing literature and does not clearly present your contributions (which are methodological in nature).

2 – Introduction:

Your literature review is comprehensive regarding foundational works, but narrow regarding current literature. I think your work would benefit if you could properly position it within current investigations in the social sciences, especially game theory approaches and the relaxation of some hypotheses, as done in Stomper (2014) or Elahi (2017), or even in opposing perspectives such as Kuhn’s non-uniqueness of algorithms [Okasha (2011)]. Also, your references are rich in logical/mathematical sources, but could be enriched with the epistemological and philosophical significance of Arrow’s impossibility theorem, given its interdisciplinary nature.

In my opinion, the language used to describe past proofs is sometimes quite harsh and should be toned down in favor of properly acknowledging their historical importance.

I think your last two paragraphs, regarding related studies, could be expanded—especially in terms of fully exploring what mechanized reasoning entails.

The main research question and methodological approach are stated in the third-to-last paragraph and should be reworked and expanded. I believe the final statement—that this proof is what Arrow himself would have sought—is quite strong and should be reconsidered. The limitations of a second-order logic proof should also be presented.

Finally, I think you should highlight that your main contribution is methodological, not theoretical, since it lacks any economic reasoning insight.

3 – Stating the impossibility theorem and translation of the premises:

As it stands, Sections 2 and 3 are the strongest parts of the paper. The balance between verbal descriptions and logical expressions is precise, and applying the same approach throughout the paper would be beneficial.

4 – Proof of the theorem

Your proof is technically sound and successfully formalizes a second-order logic proof of Arrow’s impossibility theorem. Thus, it fully addresses your research question, but I believe this section has some major issues that, if resolved, could greatly enhance the paper.

Sections 4.1–4.3, although necessary, are quite repetitive. The diagrams, in their current form, are not very helpful in sketching the proof and could be reworked. Also, the dryness of Section 4.2 hinders the understanding of the proof’s expansion. The text could accommodate more verbose descriptors. While this is a coherent choice given your overall approach, it is ultimately detrimental to the paper. I think these sections could rely more on meta-descriptors, showcasing the overall procedure instead of presenting a step-by-step proof. Since these are nested in nature, the logical pattern could be emphasized.

Additionally, although the Supplementary Material is readily available, the crux of the proofs developed there should be highlighted on pages 14 and 19.

5 – Conclusion

This section is quite succinct. It could accommodate a more thorough discussion, particularly regarding the economic insights and the limitations of the second-order logic approach. Also, it would benefit from properly placing your proof in the historical context of previous efforts and in epistemic/philosophical discussions. Since your paper may attract readers from other fields of the social sciences—due to the importance of Arrow’s impossibility theorem—you should articulate the relevance of your contribution in more general terms.

Reviewers' comments:

Reviewer's Responses to Questions

**Comments to the Author**

1. Is the manuscript technically sound, and do the data support the conclusions?

Reviewer #1: Yes

2. Has the statistical analysis been performed appropriately and rigorously?

Reviewer #1: N/A

3. Have the authors made all data underlying the findings in their manuscript fully available?

Reviewer #1: Yes

4. Is the manuscript presented in an intelligible fashion and written in standard English?

Reviewer #1: No

Reviewer #1: This paper is technically sound and has strong, albeit somewhat limited in application, conclusions. It certainly merits publication, although it needs some reworking. Please consider the following:

0 – General:

Although the language you employ is mostly precise, the text is very dry. I understand your approach of conducting a full logical proof with minimal use of verbal remarks, but it can alienate non-specialist readers. This level of abstraction works against the broader dissemination of your work. Please consider adding illustrative examples and descriptions less reliant on pure logical expressions.

1 – Abstract

Your abstract is very short. It fails to properly place your work within the existing literature and does not clearly present your contributions (which are methodological in nature).

2 – Introduction:

Your literature review is comprehensive regarding foundational works, but narrow regarding current literature. I think your work would benefit if you could properly position it within current investigations in the social sciences, especially game theory approaches and the relaxation of some hypotheses, as done in Stomper (2014) or Elahi (2017), or even in opposing perspectives such as Kuhn’s non-uniqueness of algorithms [Okasha (2011)]. Also, your references are rich in logical/mathematical sources, but could be enriched with the epistemological and philosophical significance of Arrow’s impossibility theorem, given its interdisciplinary nature.

In my opinion, the language used to describe past proofs is sometimes quite harsh and should be toned down in favor of properly acknowledging their historical importance.

I think your last two paragraphs, regarding related studies, could be expanded—especially in terms of fully exploring what mechanized reasoning entails.

The main research question and methodological approach are stated in the third-to-last paragraph and should be reworked and expanded. I believe the final statement—that this proof is what Arrow himself would have sought—is quite strong and should be reconsidered. The limitations of a second-order logic proof should also be presented.

Finally, I think you should highlight that your main contribution is methodological, not theoretical, since it lacks any economic reasoning insight.

3 – Stating the impossibility theorem and translation of the premises:

As it stands, Sections 2 and 3 are the strongest parts of the paper. The balance between verbal descriptions and logical expressions is precise, and applying the same approach throughout the paper would be beneficial.

4 – Proof of the theorem

Your proof is technically sound and successfully formalizes a second-order logic proof of Arrow’s impossibility theorem. Thus, it fully addresses your research question, but I believe this section has some major issues that, if resolved, could greatly enhance the paper.

Sections 4.1–4.3, although necessary, are quite repetitive. The diagrams, in their current form, are not very helpful in sketching the proof and could be reworked. Also, the dryness of Section 4.2 hinders the understanding of the proof’s expansion. The text could accommodate more verbose descriptors. While this is a coherent choice given your overall approach, it is ultimately detrimental to the paper. I think these sections could rely more on meta-descriptors, showcasing the overall procedure instead of presenting a step-by-step proof. Since these are nested in nature, the logical pattern could be emphasized.

Additionally, although the Supplementary Material is readily available, the crux of the proofs developed there should be highlighted on pages 14 and 19.

5 – Conclusion

This section is quite succinct. It could accommodate a more thorough discussion, particularly regarding the economic insights and the limitations of the second-order logic approach. Also, it would benefit from properly placing your proof in the historical context of previous efforts and in epistemic/philosophical discussions. Since your paper may attract readers from other fields of the social sciences—due to the importance of Arrow’s impossibility theorem—you should articulate the relevance of your contribution in more general terms.

**Do you want your identity to be public for this peer review?** For information about this choice, including consent withdrawal, please see our Privacy Policy

Reviewer #1: **Yes:** Henrique Bracarense Fagioli

---

## [Author Response · Author response to Decision Letter 1]

22 Sep 2025

Kindly refer to the "Response to Reviewers" file.

---

## [Decision Letter · Decision Letter 1]

1 Feb 2026

A full formal representation of Arrow’s impossibility theorem

PONE-D-25-11311R1

Dear Dr. Yamamoto,

We’re pleased to inform you that your manuscript has been judged scientifically suitable for publication and will be formally accepted for publication once it meets all outstanding technical requirements.

Kind regards,

Rafael Galvão de Almeida, PhD.

Academic Editor

PLOS One

Additional Editor Comments (optional):

Reviewers' comments:

Reviewer's Responses to Questions

**Comments to the Author**

Reviewer #1: All comments have been addressed

Reviewer #2: All comments have been addressed

2. Is the manuscript technically sound, and do the data support the conclusions?

Reviewer #1: Yes

Reviewer #2: Partly

3. Has the statistical analysis been performed appropriately and rigorously?

Reviewer #1: N/A

Reviewer #2: (No Response)

4. Have the authors made all data underlying the findings in their manuscript fully available?

Reviewer #1: Yes

Reviewer #2: Yes

5. Is the manuscript presented in an intelligible fashion and written in standard English?

Reviewer #1: Yes

Reviewer #2: Yes

Reviewer #1: The paper has substantially improved. However, there are still minor details to be addressed before publication. Beforehand, I thank the author for addressing all my previous comments in an attentive and corteous manner.

I ask that the author submits a clean version of the manuscript, without tracking changes. Although it doesn't impact my review, it can potentially present editorial challenges.

Also, although the author has already verified the consistency of citations, since the manuscript tracking chances has seemingly strange citation number (271 in the introduction, for an instance), I ask that the author please proceed with another verification in the clean manuscript.

Reviewer #2: (No Response)

**Do you want your identity to be public for this peer review?** For information about this choice, including consent withdrawal, please see our Privacy Policy

Reviewer #1: **Yes:** Henrique Bracarense Fagioli

Reviewer #2: **Yes:** Jincheng Zhang

---

## [Editor Report · Acceptance letter]

PONE-D-25-11311R1

PLOS One

Dear Dr. Yamamoto,

I'm pleased to inform you that your manuscript has been deemed suitable for publication in PLOS One. Congratulations! Your manuscript is now being handed over to our production team.

Kind regards,

on behalf of

Dr. Rafael Galvão de Almeida

Academic Editor

PLOS One